# Hierarchical Pretraining on Multimodal Electronic Health Records

**Xiaochen Wang[1], Junyu Luo[1], Jiaqi Wang[1], Ziyi Yin[1], Suhan Cui[1],**
**Yuan Zhong[1], Yaqing Wang[2], Fenglong Ma[1]**
[1]The Pennsylvania State University, [2]Google Research
[1]{xcwang, junyu, jqwang, ziyiyin, suhan, yuanzhong, fenglong}@psu.edu
[2]yaqingwang@google.com

## Abstract

Pretraining has proven to be a powerful technique in natural language processing (NLP), exhibiting remarkable success in various NLP downstream tasks. However, in the medical domain, existing pretrained models on electronic health records (EHR) fail to capture the hierarchical nature of EHR data, limiting their generalization capability across diverse downstream tasks using a single pretrained model. To tackle this challenge, this paper introduces a novel, general, and unified pretraining framework called MEDHMP[1], specifically designed for hierarchically multimodal EHR data. The effectiveness of the proposed MEDHMP is demonstrated through experimental results on eight downstream tasks spanning three levels. Comparisons against eighteen baselines further highlight the efficacy of our approach.

## 1 Introduction

Pretraining is a widely adopted technique in natural language processing (NLP). It entails training a model on a large dataset using unsupervised learning before fine-tuning it on a specific downstream task using a smaller labeled dataset. Pretrained models like BERT (Devlin et al., 2018) and GPT (Radford et al., 2018) have demonstrated remarkable success across a range of NLP tasks, contributing to significant advancements in various NLP benchmarks.

In the medical domain, with the increasing availability of electronic health records (EHR), researchers have attempted to pre-train domain-specific models to improve the performance of various predictive tasks further (Qiu et al., 2023). For instance, ClinicalBERT (Huang et al., 2019) and ClinicalT5 (Lehman and Johnson, 2023) are pretrained on clinical notes, and Med2Vec (Choi et al., 2016a) and MIME (Choi et al., 2018) on medical

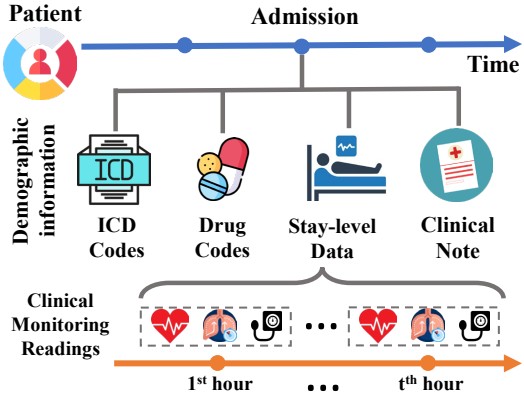

Figure 1: An illustration of EHR hierarchy.

codes. These models pretrained on a single type of data are too specific, significantly limiting their transferability. Although some pretraining models (Li et al., 2022a, 2020; Meng et al., 2021) are proposed to use multimodal EHR data[2], they ignore the heterogeneous and hierarchical characteristics of such data.

The EHR data, as depicted in Figure 1, exhibit a hierarchical structure. At the **patient** level, the EHR systems record demographic information and capture multiple admissions/visits in a time-ordered manner. Each **admission** represents a specific hospitalization period and contains multiple stay records, International Classification of Diseases (ICD) codes for billing, drug codes, and a corresponding clinical note. Each **stay** record includes hourly clinical monitoring readings like heart rate, arterial blood pressure, and respiratory rate.

In addition to the intricate hierarchy of EHR data, the **prediction tasks vary across levels**. As we move from the top to the bottom levels, the prediction tasks become more time-sensitive. Patient-level data are usually used to predict the risk of a patient suffering from potential diseases after *six months or one year*, i.e., the health risk prediction

---

[1]Source codes are available at https://github.com/XiaochenWang-PSU/MedHMP.

[2]Most of the multimodal pretraining models use medical images and other modalities, such as (Hervella et al., 2021). However, it is impossible to link EHR data and medical images in practice due to data privacy issues.

task. Admission-level data are employed for relatively shorter-term predictions, such as readmission within *30 days*. Stay-level data are typically utilized for hourly predictions, such as forecasting acute respiratory failure (ARF) within *a few hours*.

Designing an ideal "one-in-all" medical pretraining model that can effectively incorporate multimodal, heterogeneous, and hierarchical EHR data as inputs, while performing self-supervised learning across different levels, is a complex undertaking. This complexity arises due to the varying data types encountered at different levels. At the stay level, the data primarily consist of time-ordered **numerical** clinical variables. However, at the admission level, the data not only encompass sequential numerical features from stays but also include sets of **discrete** ICD and drug codes, as well as **unstructured** clinical notes. As a result, it becomes challenging to devise appropriate pretraining tasks capable of effectively extracting knowledge from the intricate EHR data.

In this paper, we present a novel **H**ierarchical **M**ultimodal **P**retraining framework (called MEDHMP) to tackle the aforementioned challenges in the **Med**ical domain. MEDHMP simultaneously incorporates **five modalities** as inputs, including patient demographics, temporal clinical features for stays, ICD codes, drug codes, and clinical notes. To effectively pretrain MEDHMP, we adopt a "bottom-to-up" approach and introduce level-specific self-supervised learning tasks. At the **stay** level, we propose reconstructing the numerical time-ordered clinical features. We devise two pretraining strategies for the **admission** level. The first focuses on modeling *intra-modality* relations by predicting a set of masked ICD and drug codes. The second involves modeling *inter-modality* relations through modality-level contrastive learning. To train the complete MEDHMP model, we utilize a two-stage training strategy from stay to admission levels[3].

We utilize two publicly available medical datasets for pretraining the proposed MEDHMP and evaluate its performance on three levels of downstream tasks. These tasks include ARF, shock and mortality predictions at the stay level, readmission prediction at the admission level, and health risk prediction at the patient level. Through our experiments, we validate the effectiveness of the proposed MEDHMP by comparing it with state-of-the-art baselines. The results obtained clearly indicate the valuable contribution of MEDHMP in the medical domain and highlight its superior performance enhancements in these predictive downstream tasks.

## 2 Methodology

As highlighted in Section 1, EHR data exhibit considerable complexity and heterogeneity. To tackle this issue, we introduce MEDHMP as a solution that leverages pretraining strategies across multiple modalities and different levels within the EHR hierarchy to achieve unification. In the following sections, we present the design details of the proposed MEDHMP.

### 2.1 Model Input

As shown in Figure 1, each patient data consist of multiple time-ordered hospital admissions, i.e., $P = [\mathcal{A}_1, \mathcal{A}_2, \cdots, \mathcal{A}_N]$, where $\mathcal{A}_i$ ($i \in [1, n]$) is the $i$-th admission, and $N$ is the number of admissions. Note that for different patients, $N$ may be different. Each patient also has a set of demographic features denoted as $\mathcal{D}$. Each admission $\mathcal{A}_i$ consists of multiple time-ordered stay-level data denoted as $\mathcal{S}_i$, a set of ICD codes denoted as $\mathcal{C}_i$, a piece of clinical notes denoted as $\mathcal{L}_i$, and a set of drug codes $\mathcal{G}_i$, i.e., $\mathcal{A}_i = \{\mathcal{S}_i, \mathcal{C}_i, \mathcal{L}_i, \mathcal{G}_i\}$. The stay-level data $\mathcal{S}_i$ contains a sequence of hourly-recorded monitoring stays, i.e., $\mathcal{S}_i = [\mathbf{S}_i^1, \mathbf{S}_i^2, \cdots, \mathbf{S}_i^{M_i}]$, where $\mathbf{S}_i^j$ represents the feature matrix of the $j$-th stay, and $M_i$ denotes the number of stays within each admission.

### 2.2 Stay-level Self-supervised Pretraining

We conduct the self-supervised pretraining in a bottom-to-top way and start from the stay level. When pretraining the stay-level data, we only use $\mathcal{S}_i$ and $\mathcal{D}$ since the diagnosis codes $\mathcal{C}_i$, drug codes $\mathcal{G}_i$ and clinical notes $\mathcal{L}_i$ are recorded at the end of the $i$-th admission. However, demographic information is highly related to a patient's clinical monitoring features in general. Due to the monitoring features being recorded with numerical values, we propose to use a reconstruction strategy as the stay-level pretraining task, as illustrated in Figure 2.

---

[3]It is important to note that we have not incorporated patient-level pertaining in MEDHMP. This decision is based on the understanding that the relations among admissions in EHR data are not as strong as consecutive words in texts. Arbitrary modeling of such relations may impede the learning of stay and admission levels.

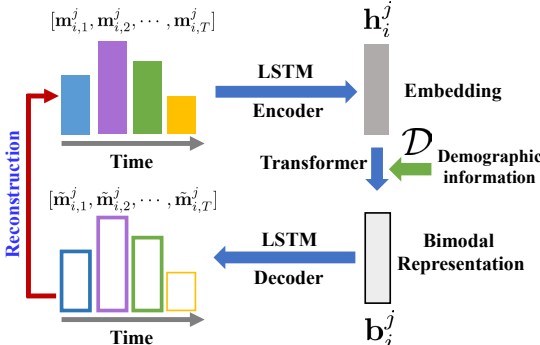

Figure 2: Stay-level self-supervised pretraining.

### 2.2.1 Stay-level Feature Encoding

Each stay $\mathbf{S}_i^j \in \mathcal{S}_i$ consists of a set of time-ordered hourly clinical features, i.e., $\mathbf{S}_i^j = [\mathbf{m}_{i,1}^j, \mathbf{m}_{i,2}^j, \cdots, \mathbf{m}_{i,T}^j]$, where $\mathbf{m}_{i,t}^j \in \mathbb{R}^{d_f}$ is the recorded feature vector at the $t$-th hour, $T$ is the number of monitoring hours, and $d_f$ denotes the number of time-series clinical features. To model the temporal characteristic of $\mathbf{S}_i^j$, we directly apply long-short term memory (LSTM) network (Hochreiter and Schmidhuber, 1997) and treat the output cell state $\mathbf{h}_i^j$ as the representation of the $j$-th stay, i.e.,

$$\mathbf{h}_i^j = \text{LSTM}_{enc}([\mathbf{m}_{i,1}^j, \mathbf{m}_{i,2}^j, \cdots, \mathbf{m}_{i,T}^j]), \quad (1)$$

where $\text{LSTM}_{enc}$ is the encoding LSTM network.

### 2.2.2 Clinical Feature Reconstruction

A naive approach to reconstructing the input stay-level feature $\mathbf{S}_i^j$ is simply applying an LSTM decoder as (Srivastava et al., 2015) does. However, this straightforward approach may not work for the clinical data. The reason is that the clinical feature vector $\mathbf{m}_{i,k}^j \in \mathbf{S}_i^j$ is *extremely sparse* due to the impossibility of monitoring all the vital signs and conducting all examinations for a patient. To accurately reconstruct such a sparse matrix, we need to use the demographic information $\mathcal{D}$ as the guidance because some examinations are highly related to age or gender, which also makes us achieve the goal of multi-modal pretraining.

Specifically, we first embed the demographic information $\mathcal{D}$ into a dense vector representation, i.e., $\mathbf{d} = \text{MLP}_d(\mathcal{D})$, where $\text{MLP}_d$ denotes the multilayer perceptron activated by the ReLU function. To fuse the demographic representation and the stay representation, we propose to use a transformer block in which self-attention is performed for modality fusion, followed by residual calculation, normalization, and a pooling operation com-

pressing the latent representation to the unified dimension size. We obtain the bimodal representation $\mathbf{b}_i^j$ as follows:

$$\hat{\mathbf{b}}_i^j = \text{Softmax}(\frac{\mathbf{W}_h^Q \langle \mathbf{h}_i^j, \mathbf{d} \rangle \cdot \mathbf{W}_h^K \langle \mathbf{h}_i^j, \mathbf{d} \rangle}{\sqrt{d_r}}) \cdot \mathbf{W}_h^V \langle \mathbf{h}_i^j, \mathbf{d} \rangle,$$

$$\mathbf{b}_i^j = \text{MaxPooling}(\text{LayerNorm}(\langle \mathbf{h}_i^j, \mathbf{d} \rangle + \hat{\mathbf{b}}_i^j)),$$
$$(2)$$

where $\langle \cdot, \cdot \rangle$ means the operation of stacking, $\mathbf{W}_h^Q$, $\mathbf{W}_h^K$, $\mathbf{W}_h^V \in \mathbb{R}^{d_r \times d_r}$ are trainable parameters, and $d_r$ is the unified size of representation.

Using the fused representation $\mathbf{b}_i^j$, MEDHMP then reconstructs the input clinical feature matrix $\mathbf{S}_i^j$. Since the clinical features are time-series data, we take $\mathbf{b}_i^j$ as the initial hidden state of the LSTM decoder $\text{LSTM}_{dec}$ to sequentially reconstruct the corresponding clinical feature $\tilde{\mathbf{m}}_{i,k}^j = \text{LSTM}_{dec}(\mathbf{b}_i^j)$.

### 2.2.3 Stay-level Pretraining Loss

After obtaining the reconstructed clinical features $[\tilde{\mathbf{m}}_{i,1}^j, \tilde{\mathbf{m}}_{i,2}^j, \cdots, \tilde{\mathbf{m}}_{i,T}^j]$, we then apply the mean squared error (MSE) as the pretraining loss to train the parameters in the stay-level as follows:

$$\mathcal{L}_{\text{stay}} = \frac{1}{N * M * T} \sum_{i=1}^N \sum_{j=1}^M \sum_{t=1}^T ||\mathbf{m}_{i,t}^j - \tilde{\mathbf{m}}_{i,t}^j||_2^2.$$
$$(3)$$

## 2.3 Admission-level Pretraining

The stay-level pretraining allows MEDHMP to acquire the sufficient capability of representing stays, laying the groundwork for the pretraining at the admission level. Next, we introduce the details of pretraining at this level.

### 2.3.1 Admission-level Feature Encoding

As introduced in Section 2.1, each admission $\mathcal{A}_i = \{\mathcal{S}_i, \mathcal{C}_i, \mathcal{L}_i, \mathcal{G}_i\}$. To conduct the self-supervised pretraining, the first step is to encode each input to a latent representation.

In Section 2.2, we can obtain the representation of each hourly feature $\mathbf{b}_i^j$ using Eq. (2). Thus, we can further have the stay-level overall representation $\mathbf{s}_i$ by aggregating all hourly representations of $\mathcal{S}_i$ via a linear transformation as follows:

$$\mathbf{s}_i = \mathbf{W}_s^\top \langle \mathbf{b}_i^1; \mathbf{b}_i^2; \cdots; \mathbf{b}_i^M \rangle + \mathbf{b}_s, \quad (4)$$

where $\langle \cdot; \cdot \rangle$ is the concatenation operation. $\mathbf{W}_s \in \mathbb{R}^{d_r \times M * d_r}$ and $\mathbf{b}_s \in \mathbb{R}^{d_r}$ are parameters.

For ICD codes $\mathcal{C}_i$ and drug codes $\mathcal{G}_i$, they will be converted to binary vectors and then map them to

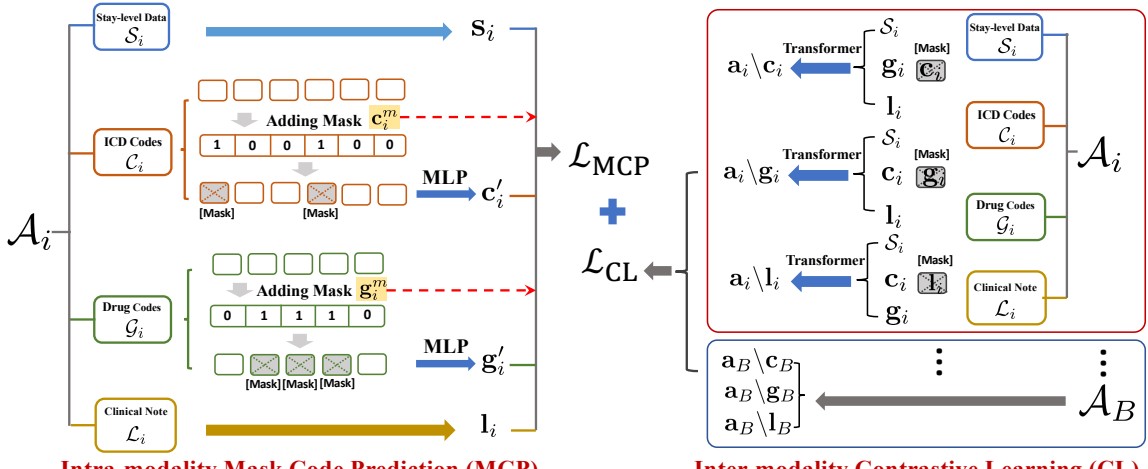

Figure 3: Admission-level self-supervised pretraining.

latent representations via MLP layers, which is similar to the mapping of the demographic information, as follows:

$$\mathbf{c}_i = \text{MLP}_c(\mathcal{C}_i), \mathbf{g}_i = \text{MLP}_g(\mathcal{G}_i). \quad (5)$$

For the unstructured clinical notes $\mathcal{L}_i$, we directly use a pretrained domain-specific encoder (Lehman and Johnson, 2023) to generate its representation $\mathbf{l}_i$.

Using the learned representations, we can conduct admission-level pretraining. Due to the unique characteristics of multimodal EHR data, we will focus on two kinds of pretraining tasks: mask code prediction for intra-modalities and contrastive learning for inter-modalities, as shown in Figure 3.

### 2.3.2 Intra-modality Mask Code Prediction

In the natural language processing (NLP) domain, mask language modeling (MLM) (Devlin et al., 2018) is a prevalent pretraining task encouraging the model to capture correlations between tokens. However, the EHR data within an admission $\mathcal{A}_i$ are significantly different from text data, where the ICD and drug codes are sets instead of sequences. Moreover, the codes are distinct. In other words, no identical codes appear in $\mathcal{C}_i$ and $\mathcal{G}_i$. Thus, it is essential to design a new loss function to predict the masked codes.

Let $\mathbf{c}_i^m \in \mathbb{R}^{|\mathcal{C}|}$ and $\mathbf{g}_i^m \in \mathbb{R}^{|\mathcal{G}|}$ denote the mask indicator vectors, where $|\mathcal{C}|$ and $|\mathcal{G}|$ denote the distinct number of ICD codes and drug codes, respectively. If the $j$-th ICD code is masked, then $\mathbf{c}_i^m[j] = 1$; otherwise, $\mathbf{c}_i^m[j] = 0$. Let $\mathbf{c}_i'$ and $\mathbf{g}_i'$ denote the embeddings learned for the remaining codes. To predict the masked codes, we need to obtain the admission representation. Toward this

end, we first stack all the learned embeddings as follows:

$$\mathbf{f}_i = \langle \mathbf{s}_i, \mathbf{c}_i', \mathbf{g}_i', \mathbf{l}_i \rangle. \quad (6)$$

Then another transformer encoder block is used to obtain the cross-modal admission representation as follows:

$$\hat{\mathbf{a}}_i = \text{Softmax}\left(\frac{\mathbf{W}_a^Q \mathbf{f}_i \cdot \mathbf{W}_a^K \mathbf{f}_i}{\sqrt{d_r}}\right) \cdot \mathbf{W}_a^V \mathbf{f}_i, \quad (7)$$
$$\mathbf{a}_i = \text{MaxPooling}(\text{LayerNorm}(\mathbf{f}_i + \hat{\mathbf{a}}_i)),$$

where $\mathbf{W}_a^Q$, $\mathbf{W}_a^K$, and $\mathbf{W}_a^V \in \mathbb{R}^{d_r \times d_r}$ are trainable parameters.

We can predict the masked codes using the learned admission representation $\mathbf{a}_i$ using Eq. (7) as follows:

$$\mathbf{p}_i^c = \text{Sigmoid}(\text{MLP}_{mc}(\mathbf{a}_i)),$$
$$\mathbf{p}_i^g = \text{Sigmoid}(\text{MLP}_{mg}(\mathbf{a}_i)), \quad (8)$$

where the predicted probability vectors $\mathbf{p}_i^c \in \mathbb{R}^{|\mathcal{C}|}$ and $\mathbf{p}_i^g \in \mathbb{R}^{|\mathcal{G}|}$.

Finally, the MSE loss serves as the objective function of the masked code prediction (MCP) task for the intra-modality modeling as follows:

$$\mathcal{L}_{\text{MCP}} = \frac{1}{N} \sum_{i=1}^{N} (||\mathbf{p}_i^c - \mathbf{c}_i^m||_2^2 + ||\mathbf{p}_i^g - \mathbf{g}_i^m||_2^2), \quad (9)$$

where $\odot$ is the element-wise multiplication.

### 2.3.3 Inter-modality Contrastive Learning

The intra-modality modeling aims to learn feature relations within a single modality using other modalities' information. On top of it, we also

consider inter-modality relations. Intuitively, the four representations $\{\mathbf{s}_i, \mathbf{c}_i, \mathbf{g}_i, \mathbf{l}_i\}$ within $\mathcal{A}_i$ share similar information. If a certain modality $\mathbf{r}_i \in \{\mathbf{s}_i, \mathbf{c}_i, \mathbf{g}_i, \mathbf{l}_i\}$ is masked, the similarity between $\mathbf{r}_i$ and the aggregated representation $\mathbf{a}_i \backslash \mathbf{r}_i$ learned from the remaining ones should be still larger than that between $\mathbf{r}_i$ and another admission's representation $\mathbf{a}_j \backslash \mathbf{r}_j$ within the same batch, where $j \neq i$.

Based on this intuition, we propose to use the noise contrastive estimation (NCE) loss as the inter-modality modeling objective as follows:

$$\mathcal{L}_{\text{CL}} = \frac{1}{3N} \sum_{i=1}^{N} \sum_{\mathbf{r}_i \in \{\mathbf{c}_i, \mathbf{g}_i, \mathbf{l}_i\}} u(\mathbf{r}_i),$$

$$u(\mathbf{r}_i) = -\log \frac{e^{\text{sim}(\mathbf{r}_i, \mathbf{a}_i \backslash \mathbf{r}_i)/\tau}}{\sum_{j=1, j\neq i}^{B} e^{\text{sim}(\mathbf{r}_i, \mathbf{a}_j \backslash \mathbf{r}_j)/\tau}}, \quad (10)$$

where $\text{sim}(\cdot, \cdot)$ denotes the cosine similarity, $B$ is the batch size, and $\tau$ is the temperature hyperparameter. $\mathbf{a}_i \backslash \mathbf{r}_i$ is obtained using Eqs. (6) and (7) by removing the masked modality $\mathbf{r}_i$. Note that in our design, $\mathbf{s}_i$ is a trained representation by optimizing the stay-level objective via Eq. (3). However, the other three modality representations are learned from scratch or the pretrained initialization. To avoid overfitting $\mathbf{s}_i$, we do not mask the stay-level representation $\mathbf{s}_i$ in Eq. (10).

### 2.3.4 Admission-level Pretraining Loss

The final loss function in the admission-level pre-training is represented as follows:

$$\mathcal{L}_{\text{admission}} = \mathcal{L}_{\text{MCP}} + \lambda \mathcal{L}_{\text{CL}}, \quad (11)$$

where $\lambda$ is a hyperparameter to balance the losses between the intra-modality mask code prediction task and the inter-modality contrastive learning.

### 2.4 Training of MEDHMP

We use a two-stage training strategy to train the proposed MEDHMP. In the first stage, we pre-train the stay-level task via Eq. (3) by convergence. In the second stage, we use the learned parameters in the first stage as initialization and then train the admission-level task via Eq. (11).

## 3 Experiments

In this section, we first introduce the data for pre-training and downstream tasks and then exhibit experimental results (*mean values of five runs*).

### 3.1 Data Extraction

We utilize two publicly available multimodal EHR datasets – MIMIC-III (Johnson et al., 2016) and MIMIC-IV (Johnson et al., 2020) – to pretrain the proposed MEDHMP. We adopt FIDDLE (Tang et al., 2020) to extract the pretraining data and use different levels' downstream tasks to evaluate the effectiveness of the proposed MEDHMP. For the stay-level evaluation, we predict whether the patient will suffer from acute respiratory failure (ARF)/shock/mortality within 48 hours by extracting data from the MIMIC-III dataset[4]. For the admission-level evaluation, we rely on the same pipeline for extracting data from the MIMIC-III dataset to predict the 30-day readmission rate. For the patient-level evaluation, we conduct four health risk prediction tasks by extracting the heart failure data from MIMIC-III following (Choi et al., 2016b) and the data of chronic obstructive pulmonary disease (COPD), amnesia, and heart failure from TriNetX[5]. The details of data extraction and statistics can be found in Appendix A. The implementation details of MEDHMP are in Appendix B.

### 3.2 Stay-level Evaluation

We conduct two experiments to validate the usefulness of the proposed MEDHMP at the stay level.

### 3.2.1 Stay-level Multimodal Evaluation

In this experiment, we take two modalities, i.e., demographics and clinical features, as the model inputs. The bimodal representation $\mathbf{b}_i^j$ learned by Eq. (2) is then fed into a fully connected layer followed by the sigmoid activation function to calculate the prediction. We use the cross entropy as the loss function to finetune MEDHMP.

We use F-LSTM (Tang et al., 2020), F-CNN (Tang et al., 2020), RAIM (Xu et al., 2018), and DCMN (Feng et al., 2019) as the baselines. The details of each baseline can be found in Appendix C. We utilize the Area Under the Receiver Operating Characteristic curve (AUROC) and the Area Under the Precision-Recall curve (AUPR) as evaluation metrics.

The experimental results are presented in Table 1, showcasing the superior performance of MEDHMP compared to the bimodal baselines in all three stay-level tasks. This indicates the proficiency of MEDHMP in effectively utilizing both

---

[4]https://github.com/MLD3/FIDDLE-experiments/tree/master/mimic3_experiments

[5]https://trinetx.com/

| Task | ARF | | Shock | | Mortality | |
|---|---|---|---|---|---|---|
| Metric | AUROC | AUPR | AUROC | AUPR | AUROC | AUPR |
| F-LSTM | 69.67 | 10.57 | 70.28 | 23.09 | 81.55 | **48.62** |
| F-CNN | 69.61 | 10.68 | 69.27 | 23.51 | 80.71 | 42.29 |
| RAIM | 59.38 | 8.42 | 66.20 | 20.02 | 77.17 | 39.96 |
| DCMN | 68.98 | 10.07 | 68.68 | 21.72 | 80.05 | 42.93 |
| MEDHMP | **71.66** | **14.34** | **71.04** | **24.19** | **82.17** | 47.52 |

Table 1: Results (%) on stay-level tasks.

clinical and demographic features. Remarkably, MEDHMP demonstrates a particularly strong advantage when handling tasks with smaller-sized datasets (See Table 8 for data scale). This observation suggests that MEDHMP greatly benefits from our effective pre-training procedure, enabling it to deliver impressive performance, especially in low-resource conditions.

Note that in the previous work (Yang and Wu, 2021), except for the demographics and clinical features, clinical notes are used to make predictions on the ARF task. We also conducted such experiments on the three tasks, and the results are listed in Appendix D. The experimental results still demonstrate the effectiveness of the proposed pretraining framework.

### 3.2.2 Stay-level Unimodal Evaluation

To validate the transferability of the proposed MEDHMP, we also conduct the following experiment by initializing the encoders of baselines using the pretrained MEDHMP. In this experiment, we only take the clinical features as models' inputs. Two baselines are used: LSTM (Hochreiter and Schmidhuber, 1997) and Transformer (Vaswani et al., 2017). We use the pretrained LSTM encoder $\text{LSTM}_{enc}$ in Section 2.2.1 to replace the original linear encoders in LSTM and Transformer. Our encoder will be finetuned with the training of LSTM and Transformer.

The experimental results on the ARF task are shown in Figure 4. As mentioned in Section 2.4, we train the LSTM encoder $\text{LSTM}_{enc}$ twice. "w. $\text{MEDHMP}_a$" means that the baselines use a well-trained **admission**-level $\text{LSTM}_{enc}$. "w. $\text{MEDHMP}_s$" indicates that the baselines use a **stay**-level trained $\text{LSTM}_{enc}$. "Original" denotes the original baselines. We can observe that using partially- or well-trained encoders helps improve performance. These results also confirm the necessity of the proposed two-stage training strategy.

### 3.3 Admission-level Evaluation

We also adopt the readmission prediction task within 30 days to evaluate MEDHMP at the ad-

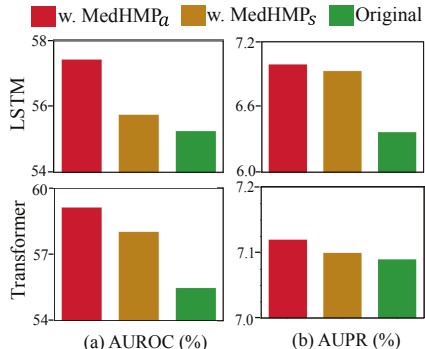

Figure 4: Unimodal evaluation on the ARF task.

| Model | AUROC | AUPR |
|---|---|---|
| BertLstm | 63.35 | 7.24 |
| LstmBert | 60.67 | 6.84 |
| BertCnn | 63.07 | 7.19 |
| CnnBert | 61.59 | 7.04 |
| BertStar | 61.28 | 6.84 |
| StarBert | 60.67 | 6.84 |
| BertEncoder | 61.94 | 6.82 |
| EncoderBert | 60.57 | 7.00 |
| MEDHMP | **67.77** | **9.34** |

Table 2: Results (%) on the readmission task.

mission level. In this task, the model will task all modalities as the input, including demographics, clinical features, ICD codes, drug codes, and a corresponding clinical note for admission. In this experiment, we first learn the representations, i.e., $\mathbf{s}_i$ using Eq. (4), $\mathbf{c}_i$ and $\mathbf{g}_i$ via Eq. (5), and $\mathbf{l}_i$, to obtain the stacked embedding $\mathbf{f}_i$. We then apply Eq. (7) to obtain the admission embedding $\mathbf{a}_i$. Finally, a fully connected layer with the sigmoid function is used for prediction. We still use the cross-entropy loss as the optimization function.

We follow the existing work (Yang and Wu, 2021) and use its **eight** multimodal approaches as baselines, which adopt modality-specific encoders and perform modality aggregation via a gating mechanism. Different from the original model design, we perform a pooling operation on the latent representation of multiple clinical time series belonging to a specific admission, such that baselines can also take advantage of multiple stays. Details of these models can be found in Appendix D. We still use AUROC and AUPR as evaluation metrics.

Admission-level results are listed in Table 2, and we can observe that the proposed MEDHMP outperforms all baselines. Compared to the best baseline performance, the AUROC and AUPR scores of MEDHMP increase 7% and 29%, respectively. These results once again prove the effectiveness of the proposed pretraining model.

| Database | MIMIC-III | | | TriNetX | | | | | | | | |
|---|---|---|---|---|---|---|---|---|---|---|---|---|
| Task | Heart Failure | | | Heart Failure | | | COPD | | | Amnesia | | |
| Metric | AUPR | F1 | KAPPA | AUPR | F1 | KAPPA | AUPR | F1 | KAPPA | AUPR | F1 | KAPPA |
| $LSTM_a$ | **57.83** | **59.40** | **35.86** | **50.16** | **46.08** | **29.26** | **50.16** | **49.34** | **34.64** | **48.68** | **49.64** | **34.46** |
| LSTM | 57.83 | 56.70 | 33.03 | 48.20 | 44.44 | 26.64 | 49.52 | 47.76 | 33.44 | 47.92 | 48.80 | 32.98 |
| $Dipole_a$ | **59.71** | **60.50** | **37.68** | **47.70** | **41.86** | **25.52** | 48.92 | **41.06** | **28.30** | **48.74** | **45.78** | **30.78** |
| Dipole | 59.43 | 58.63 | 36.03 | 47.16 | 40.16 | 24.28 | **49.44** | 39.48 | 27.86 | 48.36 | 45.63 | 30.40 |
| $RETAIN_a$ | **68.71** | **66.20** | **47.12** | **58.16** | **52.18** | **35.64** | **57.62** | **50.66** | **38.36** | **62.70** | **56.50** | **43.90** |
| RETAIN | 67.76 | 65.56 | 45.63 | 57.50 | 50.88 | 34.52 | 57.40 | 49.85 | 37.36 | 62.52 | 56.32 | 43.66 |
| $AdaCare_a$ | 58.40 | **59.47** | 35.77 | **57.63** | **47.98** | **32.03** | 54.06 | **47.10** | **34.70** | **62.62** | **52.56** | **41.54** |
| AdaCare | **59.40** | 57.58 | **35.84** | 55.43 | 45.13 | 31.43 | **56.63** | 46.60 | 34.53 | 61.62 | 50.54 | 39.22 |
| $HiTANet_a$ | 69.42 | **68.44** | **50.01** | **60.12** | **50.48** | **36.08** | **64.04** | **54.46** | **43.38** | **67.54** | **58.18** | **47.78** |
| HiTANet | **70.36** | 66.60 | 46.60 | 54.76 | 47.92 | 32.04 | 60.10 | 52.40 | 39.93 | 63.08 | 54.60 | 43.44 |

Table 3: Performance (%) of baselines with/without pretraining for the health risk prediction task.

## 3.4 Patient-level Evaluation

Even though MEDHMP has not been pretrained on patient-level tasks, it is still capable of handling tasks at this level since its unimodal encoders acquire the ability to generate a high-quality representation of each admission, thus become feasible to be utilized to boost existing time series-targeting models. Health risk prediction, which utilizes a sequence of hospital admissions for illness forecasting, is applied as the task at the patient level.

In this experiment, the model will take a sequence of admission-level ICD codes as the input, which is still a unimodal evaluation. We use the following approaches as baselines: LSTM (Hochreiter and Schmidhuber, 1997), Dipole (Ma et al., 2017), RETAIN (Choi et al., 2016b), AdaCare (Ma et al., 2020), and HiTANet (Luo et al., 2020). Details of these approaches can be found in Appendix E. Following previous health risk prediction work (Chen et al., 2021; Cui et al., 2022a), we use AUPR, F1, and Cohen's Kappa as the evaluation metrics.

### 3.4.1 Performance Comparison

The experimental results are shown in Table 3, where the approach with the subscript "$a$" denotes the baseline using the pretrained MEDHMP to initialize the ICD code embedding $c_i$ via Eq. (5). We can find that introducing the pretrained unimodal encoder from MEDHMP achieves stable improvement across most of the baselines and tasks. These results demonstrate the flexibility and effectiveness of our proposed MEDHMP in diverse medical scenarios. The knowledge from our pretrained model can be easily adapted to any sub-modality setting.

### 3.4.2 Influence of Training Size

Intuitively, pretraining could lead to improved initialization performance compared to models trained from scratch, thereby enhancing its suitability in

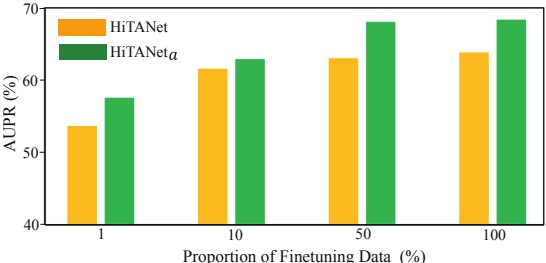

Figure 5: Performance change with different training data sizes using HiTANet on the TriNetX amnesia prediction task.

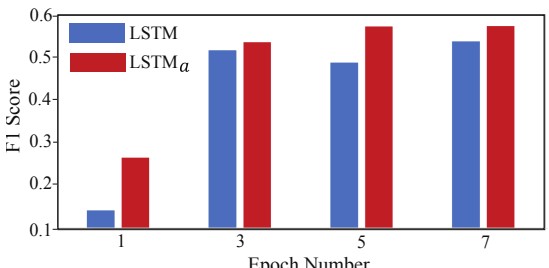

Figure 6: Performance changes w.r.t. the number of epochs on the TriNetX heart failure prediction task.

low-resource settings such as zero-shot learning and few-shot learning. Inspired by these characteristics, we explore low-resource settings that simulate common real-world health-related scenarios. We replicate the experiments introduced in the previous section but vary the size of the training set from 1% to 100%.

Figure 5 shows the experimental results using the HiTANet model. We can observe that using the pretraining initialization, $HiTANet_a$ always achieves better performance. Even with 10% training data, it can achieve comparable performance with the plain HiTANet using 100% data. This promising result confirms that the proposed pretraining framework MEDHMP is useful and meaningful for medical tasks, especially when the training data are insufficient.

| Database | MIMIC-III | | | TriNetX | | | | | | | | |
|---|---|---|---|---|---|---|---|---|---|---|---|---|
| Task | Heart Failure | | | Heart Failure | | | COPD | | | Amnesia | | |
| Metric | AUPR | F1 | KAPPA | AUPR | F1 | KAPPA | AUPR | F1 | KAPPA | AUPR | F1 | KAPPA |
| LSTM$_{a+s}$ | **57.83** | **59.40** | **35.86** | **50.16** | **46.08** | **29.26** | **50.16** | **49.34** | **34.64** | **48.68** | **49.64** | **34.46** |
| LSTM$_a$ | 57.57 | 58.27 | 35.67 | 49.88 | 44.86 | 28.58 | 49.90 | 47.65 | 33.77 | 48.48 | 48.70 | 33.52 |
| LSTM | **57.83** | 56.70 | 33.03 | 48.20 | 44.44 | 26.64 | 49.52 | 47.76 | 33.44 | 47.92 | 48.80 | 32.98 |

Table 4: Performance (%) of LSTM on the health risk prediction task, which is initialized with parameters from MEDHMP, admission-level pertaining only, and without pretraining.

### 3.4.3 Convergence Analysis with Pretraining

In this experiment, we aim to explore whether using pretraining can speed up the convergence of model training. We use the basic LSTM model as the baseline and output the testing performance at each epoch. Figure 6 shows the results. We can observe that at each epoch, the F1 score of LSTM$_a$ is higher than that of LSTM, indicating the benefit of using pretraining. Besides, LSTM$_a$ achieves the best performance at the 5-th epoch, but the F1 score of the plain LSTM still vibrates. Thus, these results clearly demonstrate that using pretraining techniques can make the model converge faster with less time and achieve better performance.

## 4 Ablation Study

### 4.1 Hierarchical Pretraining

For the comprehensive analysis of the effect of stay- and admission-level pretraining, we perform ablation studies spanning downstream tasks at all three levels. Results of patient-level, admission-level, and stay-level tasks are listed in Table 4, 5 and 6, respectively. The subscripts "a" (admission) and "s" (stay) in these tables indicate which pretrained model is used as the initialization of MEDHMP.

From the results of all three tables, we can observe that the combination of both stay- and admission-level pretraining manifests superior performance, further underlining the necessity of adopting hierarchical pretraining strategies. Besides, compared with the model without any pretraining techniques, merely using a part of the proposed pretraining strategy for initialization can improve the performance. These observations imply the correct rationale behind our design of hierarchical pretraining strategies.

### 4.2 Multimodal Modeling

To investigate how intra- and inter-modality modeling techniques benefit our admission-level pretraining, we perform an ablation study on three tasks at the stay-level to examine the effectiveness of Mask

| Model | AUROC | AUPR |
|---|---|---|
| MEDHMP$_{a+s}$ | **67.77** | **9.34** |
| MEDHMP$_a$ | 65.75 | 9.08 |
| MEDHMP$_s$ | 64.87 | 8.60 |
| MEDHMP | 64.74 | 8.61 |

Table 5: Results (%) on the readmission task, where MEDHMP is initialized with bi-level pretraining, admission-level pretraining, stay-level pertaining, and without pretraining.

| Task | ARF | | Shock | | Mortality | |
|---|---|---|---|---|---|---|
| Metric | AUROC | AUPR | AUROC | AUPR | AUROC | AUPR |
| MEDHMP$_{a+s}$ | **71.66** | **14.34** | **71.04** | **24.19** | **82.17** | **47.52** |
| MEDHMP$_s$ | 64.65 | 10.59 | 67.94 | 22.50 | 79.67 | 42.66 |
| MEDHMP | 64.06 | 10.80 | 67.71 | 23.19 | 79.04 | 40.12 |

Table 6: Results (%) on the stay-level task, where MEDHMP is initialized with bi-level pretraining, stay-level pertaining, and without pretraining.

Code Prediction (MCP) and Contrastive Learning (CL) losses. We compare MEDHMP pretrained with all loss terms, with MCP and stay-level loss, with CL and stay-level loss, and stay-level loss only, respectively. Results presented in Table 7 clearly demonstrate the efficacy of each proposed loss term as well as the designed pretraining strategy. Besides, lacking each of them results in performance reduction, highlighting that combining intra- and inter-modality modeling is indispensable for boosting the model comprehensively.

## 5 Related Work

Predictive modeling using EHR data has attracted significant attention in recent years (Cui et al., 2022b; Ma et al., 2021; Xiao et al., 2018; Wang et al., 2022). To enhance predictive performance, pretraining techniques have been explored. In this section, we provide a concise overview of studies conducted on pretraining with both single-modal and multimodal EHR data.

### 5.1 Unimodal Pretraining with EHR Data

Several pretrained models have been proposed by utilizing single-modal EHR data. Building upon the success of Large Language Models

| Task | ARF | | Shock | | Mortality | |
|---|---|---|---|---|---|---|
| Metric | AUROC | AUPR | AUROC | AUPR | AUROC | AUPR |
| MEDHMP$_{a+s}$ | **71.66** | **14.34** | **71.04** | 24.19 | **82.17** | **47.52** |
| MEDHMP$_{MCP+s}$ | 64.91 | 12.35 | 68.61 | **25.29** | 81.32 | 47.50 |
| MEDHMP$_{CL+s}$ | 62.99 | 13.88 | 70.05 | 22.81 | 80.58 | 44.40 |
| MEDHMP$_s$ | 64.65 | 10.59 | 67.94 | 22.50 | 79.67 | 42.66 |

Table 7: Ablation results (%) regarding MCP and CL on the readmission task.

(LLMs) (Devlin et al., 2018; Radford et al., 2018) in NLP, researchers have endeavored to train medical-specific language models using **clinical notes** (Li et al., 2022b; Lehman and Johnson, 2023; Alsentzer et al., 2019; Peng et al., 2019) and PubMed data (Luo et al., 2022; Lee et al., 2020; Yuan et al., 2022; Jin et al., 2019; Warikoo et al., 2021). However, these models primarily rely on mask language modeling techniques for pretraining, thereby overlooking the distinctive characteristics of medical data.

Given the time-ordered nature of admissions, **medical codes** can be treated as a sequence. Some pertaining models have proposed to establish representations of medical codes (Rasmy et al., 2021; Li et al., 2020; Shang et al., 2019; Choi et al., 2016a, 2018). Nevertheless, these studies still adhere to the commonly used pretraining techniques in the NLP domain. Another line of work (Tipirneni and Reddy, 2022; Wickstrøm et al., 2022) is to conduct self-supervised learning on **clinical features**. However, these pretrained models can only be used for the downstream tasks at the stay level, limiting their transferability in many clinical application scenarios.

### 5.2 Multimodal Pretraining with EHR data

Most of the multimodal pretraining models in the medical domain are mainly using medical images (Qiu et al., 2023) with other types of modalities, such as text (Hervella et al., 2021, 2022a,b; Khare et al., 2021) and tabular information (Hager et al., 2023). Only a few studies focus on pretraining on multimodal EHR data without leveraging medical images. The work (Li et al., 2022a, 2020) claims their success on multimodal pretraining utilizing numerical clinical features and diagnosis codes. In (Liu et al., 2022), the authors aim to model the interactions between clinical language and clinical codes. Besides, the authors in (Meng et al., 2021) use ICD codes, demographics, and topics learned from text data as the input and utilize the mask language modeling technique to pretrain the model. However, all existing pretrained work on EHR data still follows the routine of NLP pretraining but ignores the hierarchical nature of EHRs in their pretraining, resulting in the disadvantage that the pretrained models cannot tackle diverse downstream tasks at different levels.

## 6 Conclusion

In this paper, we present a novel pretraining model called MEDHMP designed to address the hierarchical nature of multimodal electronic health record (EHR) data. Our approach involves pretraining MEDHMP at two levels: the stay level and the admission level. At the stay level, MEDHMP uses a reconstruction loss applied to the clinical features as the objective. At the admission level, we propose two losses. The first loss aims to model intra-modality relations by predicting masked medical codes. The second loss focuses on capturing inter-modality relations through modality-level contrastive learning. Through extensive multimodal evaluation on diverse downstream tasks at different levels, we demonstrate the significant effectiveness of MEDHMP. Furthermore, experimental results on unimodal evaluation highlight its applicability in low-resource clinical settings and its ability to accelerate convergence.

## 7 Limitations

Despite the advantages outlined in the preceding sections, it is important to note that MEDHMP does have its limitations. Owing to the adoption of a large batch size to enhance contrastive learning (see Appendix B for more details), it becomes computationally unfeasible to fine-tune the language model acting as the encoder for clinical notes during our admission-level pretraining. As a result, ClinicalT5 is held static to generate a fixed representation of the clinical note, which may circumscribe potential advancements. Additionally, as described in Appendix A, we only select admissions with ICD-9 diagnosis codes while excluding those with ICD-10 to prevent conflicts arising from differing coding standards. This selection process, however, implies that MEDHMP currently lacks the capacity to be applied in clinical scenarios where ICD-10 is the standard for diagnosis code.

### Acknowledgement

This work is partially supported by the US National Science Foundation under Grants #2238275 and #2212323, and the US National Institutes of Health under Grant R01AG077016.

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

## A  Data Processing

We utilize two publicly available multimodal EHR datasets – MIMIC-III (Johnson et al., 2016) and MIMIC-IV (Johnson et al., 2020) – to pretrain the proposed MEDHMP. Considering that MIMIC-III uses ICD-9 codes while MIMIC-IV incorporates both ICD-9 and ICD-10 codes, we only select admissions with ICD-9 diagnosis codes to avoid potential conflicts between different coding standards. To prevent the label leakage issue during the testing stage, we remove all signals related to downstream tasks from the original data.

We adopt the EHR-oriented preprocessing pipeline, FIDDLE (Tang et al., 2020), for feature and label extraction at the stay level. We standardize the length of the clinical monitoring feature to $T = 48$ hours, which is the upper bound for clinical feature-related tasks mentioned in (Tang et al., 2020). We filter out the features with a frequency lower than 5% since extremely sparse features can significantly harm computing efficiency and be memory burdensome. After data preprocessing, each hourly clinical feature $\mathbf{m}_{i,t}^j$ is represented as a 1,318-dimensional sparse vector, i.e., $d_f = 1,318$. The demographics for each patient are represented as a 73-dimensional sparse vector, i.e., the length of $\mathcal{D}$ is 73. The number of unique ICD codes $|\mathcal{C}|$

is 7,686, and the number of unique drug codes $|\mathcal{G}|$ is 1,701. Finally, we get 99,000 admissions with 100,563 stays for pretraining MEDHMP.

The three datasets extracted from TriNetX are supervised by clinicians. We employ the extraction method described in (Choi et al., 2016b) to identify case patients. Specifically, we identify the initial diagnosis date and utilize the patient's historical data leading up to a six-month window, where the diagnosis date marks its end. This approach ensures that we prevent label leakage and successfully accomplish the objective of early prediction. Three control cases are chosen for each positive case based on matching criteria such as gender, age, race, and underlying diseases. For control patients, we use the last 50 visits in the database.

The statistics of data used for both pretraining and downstream tasks can be found in Table 8.

## B  Implementation and Configuration

All models were implemented using PyTorch 2.0.0 and Python 3.9.12. Preprocessing and experiments were conducted in the Ubuntu 20.04 system with 376 GB of RAM and two NVIDIA A100 GPUs.

Each experiment was repeated five times to eliminate randomness, and the mean of the evaluation metrics was reported in all experimental results.

For unimodal evaluations, we used the same set of hyperparameters, regardless of whether a pretrained encoder was used, to ensure a fair comparison. For multimodal evaluations, we either used the hyperparameters reported by the authors of the baselines or suggested in their release codes. For detailed hyperparameters not provided by these authors, we used the same hyperparameters as in our model for a fair comparison.

$d_r$ was set to 256 for our pretraining and evaluation in downstream tasks. For stay-level pretraining, our model was pretrained for 200 epochs with a learning rate of 5e-4, a batch size of 128, and a weight decay of 1e-8. At the admission level, our model was pretrained for 300 epochs, with a learning rate of 2e-5 and a weight decay of 1e-8. Following previous works (Chen et al., 2022, 2020), which emphasized the necessity of adopting a large batch size in contrastive learning, we set the batch size to 4096 to enhance our inter-modality modeling. $\tau$ in contrastive learning loss was set to 0.1. The hyperparameter $\lambda$ mentioned in Eq. (11) was set to 0.1 to balance the masked code prediction (MCP) and contrastive learning (CL) losses

| Pretraining | Number of of stays | | | 100,563 | | |
| | Number of admissions | | | 99,000 | | |

| | Level | Dataset | Predictive Task | Total | Positive | Negative |
|---|---|---|---|---|---|---|
| Downstream | Stay | MIMIC-III | ARF within 48 hours | 5,038 | 402 | 4,636 |
| | | | Shock within 48 hours | 7,182 | 693 | 6,489 |
| | | | Mortality within 48 hours | 11,695 | 1,581 | 10,114 |
| | Admission | MIMIC-III | Readmission within 30 days | 33,179 | 1,444 | 31,735 |
| | Patient | MIMIC-III | Heart Failure after six months | 7,522 | 2,820 | 4,702 |
| | | TriNetX | COPD after six months | 29,256 | 7,314 | 21,942 |
| | | | Amnesia after six months | 11,928 | 2,982 | 8,946 |
| | | | Heart Failure after six months | 12,320 | 3,080 | 9,240 |

Table 8: Data statistics.

| Model | Main Modality | Auxiliary Modalities | Clinical Feature Encoder |
|---|---|---|---|
| BertLstm | Clinical Notes | Clinical Features and Demographics | LSTM |
| LstmBert | Clinical Features | Clinical Notes and Demographics | LSTM |
| BertCnn | Clinical Notes | Clinical Features and Demographics | CNN |
| CnnBert | Clinical Features | Clinical Notes and Demographics | CNN |
| BertStar | Clinical Notes | Clinical Features and Demographics | StarTransformer |
| StarBert | Clinical Features | Clinical Notes and Demographics | StarTransformer |
| BertEncoder | Clinical Notes | Clinical Features and Demographics | Transformer |
| EncoderBert | Clinical Features | Clinical Notes and Demographics | Transformer |

Table 9: Baselines for the admission-level task.

in the stay-level pretraining. The masking rate in the MCP task was set to 15%, following the design of (Devlin et al., 2018). The optimizer used throughout the pretraining stage was AdamW.

For downstream tasks, we selected the hyperparameters of our model using Grid Search. The batch size was chosen from the set [16, 32, 64], and the learning rate was searched in the range from 2e-5 to 5e-3. The maximum number of epochs was set to 30, and the patience for early stopping and weight decay were configured to 5 and 1e-2, respectively, to avoid overfitting. We found that the SGD optimizer performed better during the fine-tuning procedure.

## C  Stay-level Experiments

Besides unimodal baselines mentioned in Section 3.2.2, the following approaches serving as baselines in the multimodal evaluation at the stay level are listed below: (1) **F-LSTM** (Tang et al., 2020) is a classic Long Short-Term Memory (LSTM) model taking concatenation of clinical features and demographic information as input. (2) **F-CNN** (Tang et al., 2020) is a typical Convolutional Neural Network (CNN) architecture using the concatenation of clinical features and demographic information

for prediction. (3) **Raim** (Xu et al., 2018) is an attention-based model specially designed for analyzing ICU monitoring data, which uses a combination of attention mechanisms and multimodal data integration. (4) **DCMN** (Feng et al., 2019) combines two separate memory networks, one for processing clinical time series and one for processing static tables. Its dual-attention mechanism design allows the model to aggregate features effectively.

## D  Stay-level Experiments with Clinical Notes

All the baselines utilized in the readmission prediction task are based on the previous work (Yang and Wu, 2021). In their study, the authors investigate various combinations of unimodal encoders and employ a gating mechanism for modality aggregation. In this approach, one modality is considered the main modality, and the embeddings from the other modalities are added as auxiliary modalities. Specific details regarding the composition of these baselines, including how the unimodal encoders are combined, can be found in Table 9.

The experimental results are presented in Table 10. It is evident that relying solely on a single modality, such as clinical notes, is inadequate for

| Modalities | Models | ARF | | Shock | | Mortality | |
|---|---|---|---|---|---|---|---|
| | | AUROC | AUPR | AUROC | AUPR | AUROC | AUPR |
| Clinical Notes | ClinicalT5 (Lehman and Johnson, 2023) | 50.06 | 5.92 | 56.76 | 13.01 | 72.18 | 26.55 |
| | ClinicalBERT (Huang et al., 2019) | 51.75 | 7.60 | 44.28 | 9.89 | 58.17 | 16.42 |
| Demographics + Clinical Features + Clincial Notes | BertLstm (Yang and Wu, 2021) | 64.90 | 9.03 | 70.22 | 23.32 | 80.80 | **45.59** |
| | LstmBert (Yang and Wu, 2021) | 61.74 | 9.79 | 66.31 | 22.56 | 78.31 | 42.31 |
| | BertCnn (Yang and Wu, 2021) | 67.04 | 9.49 | 64.34 | 21.13 | 81.12 | 44.01 |
| | CnnBert (Yang and Wu, 2021) | 63.38 | 10.31 | 66.40 | 22.46 | 77.25 | 36.43 |
| | BertStar (Yang and Wu, 2021) | 58.11 | 6.86 | 59.24 | 19.58 | 76.24 | 36.12 |
| | StarBert (Yang and Wu, 2021) | 51.96 | 5.73 | 58.89 | 16.92 | 76.19 | 34.87 |
| | BertEncoder (Yang and Wu, 2021) | 61.30 | 9.06 | 52.88 | 14.66 | 74.95 | 35.02 |
| | EncoderBert (Yang and Wu, 2021) | 62.65 | 7.44 | 61.39 | 18.66 | 74.35 | 33.68 |
| | MEDHMP (ours) | **71.67** | **11.05** | **70.57** | **24.30** | **82.06** | 42.18 |

Table 10: Comparison results (%) of stay-level tasks using clinical notes.

achieving accurate predictions when compared to multimodal baselines. Among all the multimodal models, our proposed MEDHMP consistently outperforms the others in the majority of scenarios. These results highlight two key findings: (1) the significance of integrating multimodal information in health predictive modeling tasks and (2) the efficacy of the proposed pretraining technique.

## E Patient-level Experiments

Baselines regarding the patient-level task are listed below. (1) **LSTM**(Hochreiter and Schmidhuber, 1997) is a typical backbone model appearing in time series forecasting tasks. (2) **HiTANet**(Luo et al., 2020) adopts the time-aware attention mechanism design that enables itself to capture the dynamic disease progression pattern. (3) **Dipole**(Ma et al., 2017) relies on the combination of bidirectional GRU and attention mechanism to analyze sequential visits of a patient. (4) **AdaCare**(Ma et al., 2020) applies the Convolutional Neural Network for feature extraction, followed by a GRU block for prediction. (5) **Retain** (Choi et al., 2016b) utilizes the reverse time attention mechanism to capture dependency between various visits of a patient.

## F Evaluation Metrics

Evaluation metrics used in our experiments are listed below:

- **AUROC** (Area Under the Receiver Operating Characteristic Curve) represents the likelihood that a classifier will rank a randomly chosen positive instance higher than a randomly chosen negative instance. It provides an aggregate measure of performance across all possible classification thresholds.

| Task | Mortality | |
|---|---|---|
| Metric | AUROC | AUPR |
| F-LSTM | 82.30 | 45.01 |
| F-CNN | 76.37 | 35.11 |
| RAIM | 83.64 | 46.40 |
| DCMN | 83.57 | 46.96 |
| MEDHMP | **84.43** | **49.00** |

Table 11: Results (%) on stay-level tasks.

- **AUPRC** (Area Under the Precision-Recall Curve) measures the area beneath the Precision-Recall curve, a plot of the precision against recall for different threshold values.

- **F1 Score** is the harmonic mean of precision and recall, offering a balance between the two when their values diverge.

- **Cohen's Kappa** is a statistic that measures inter-rater agreement for categorical items, accounting for the possibility of the agreement occurring by chance.

## G Experiments on EICU Database

To further validate the transferability of our proposed MEDHMP, we conduct experiments using data from additional medical databases, i.e., eICU[6]. Results can be found in Table 11. Our proposed MEDHMP shows superior performance consistent with experiments on the MIMIC-III database, implying its excellent capability of learning general medical features.

---

[6]https://eicu-crd.mit.edu/