# OpenReview forum: "Hierarchical Pretraining on Multimodal Electronic Health Records"
_EMNLP/2023/Conference — EMNLP 2023 Main_

### Official Review · Reviewer_B8GM · 2023-08-05

**Soundness:** 4

**Excitement:**

4: Strong: This paper deepens the understanding of some phenomenon or lowers the barriers to an existing research direction.

**Paper Topic And Main Contributions:**

This paper develops an approach to pretraining model and method tailored specifically to the hierarchy of electronic health records. This paper includes 3 different levels of modalities: patient-level demographics, admission-level modalities (discharge summaries, ICD and Drug codes) and stay-level clinical monitoring readings. They use pretraining at the two lower levels of the hierarchy (admission- and stay-level), using an intra-modality reconstruction loss at the stay-level, an intra-modality masked code prediction loss at the admission level, and a inter-modality contrastive loss at the admission level (by pooling the stay-level features into one admission-level feature vector).

**Questions For The Authors:**

Why use an LSTM rather than a transformer for the stay-level feature encoding?

Can you ablate for admission-level training only (see above)?

Can you explain section 2.3.2? I am unclear how equation 9 works.

Can you explain section 3.2.2? It is confusing how you modify the Transformer baseline with LSTM_enc because you can't warm-start the Transformer from a pre-trained LSTM.

**Reasons To Accept:**

1. Great figures, mostly very clear and well-written.
2. Almost all of the reasonable experiments were done, and the results across multiple datasets and tasks are convincing. It is especially nice to note that the model performs well on smaller datasets, indicating it might generally be better for low-resource settings.
3. The hierarchical pre-training is a valuable tool for the many tasks that could benefit from data at the stay level as well as clinical notes and ICD codes. If open-sourced, this model would be broadly helpful to anyone doing machine learning with electronic health records.


**Reasons To Reject:**

1. They acknowledge that there is no patient-level pre-training objective, only stay and admission level ones. The authors point out that this might impede the other training objectives, but seeing as so many health-related tasks (e.g. diagnosis, readmission prediction) could benefit from learning interactions across multiple admissions, this seems like a missing piece.
2. Some sections are unclear (2.3.2, equation 9 especially and section 3.2.2).
3. Slightly limited novelty: There has been a fair amount of work on pertaining and even some hierarchical pretraining. However, they do a great job citing related works, and their inclusion of the most fine-grained data, the stay-level data, is novel in multimodal pretraining.

More Minor:

4. Unclear why they use an LSTM rather than a transformer for the stay-level feature encoding. h_i^j could easily also be the CLS token of a transformer, especially given that some experiments were done with a transformer (figure 4) and seemed to do better than the LSTM. I may be misunderstanding this though.
5. Lacking an ablation for admission-level only training (i.e. only using the clinical notes and ICD codes and not the stay-level clinical data). What does the stay-level clinical data bring to the table for admission and patient-level tasks both at inference and during training?

**Reproducibility:**

4: Could mostly reproduce the results, but there may be some variation because of sample variance or minor variations in their interpretation of the protocol or method.

**Reviewer Confidence:**

4: Quite sure. I tried to check the important points carefully. It's unlikely, though conceivable, that I missed something that should affect my ratings.

**Typos Grammar Style And Presentation Improvements:**

Line 329: I think it should be equation 11 instead of 6.

Table 2 caption: of -> on

Line 383, what are the sizes of the datasets?

Can you clarify why you chose a two-stage training rather than mixing all the objectives? I assume this might be because the stay-level objective of reconstruction is “harder” and the admission-level one might mess it up, but clarification would be much appreciated.

Section 3.2.2 is confusing, LSTM_enc replaces the linear encoders in LSTM and Transformer?

Equation 7: typo, f needs an i

Equation 9: I think there is a typo because the optimal solution here is that p is always 1.

It would be great to have a visualization of what multimodal training does. Maybe some tsne figures when ablating over the training objectives colored by classes? How do the admission-level or stay-level representations change based on if the training is multimodal or not?

---

> ### Author Rebuttal · Authors · 2023-08-29
>
> The comments provided by the reviewer  have been greatly beneficial and we truly appreciate it. We are more than willing to address each question respectively for the clarification.
>
> `>>Question 1:` __*Lack of explanation of using LSTM rather than Transformer*__
>
> `>>Response:` Thanks for noticing this detail. We had considered using Transformer instead of LSTM when designing our architecture. However, the experimental results listed in Table 2 and Table 6 show that baselines using transformer as the encoder of stay-level features usually manifest less competitive performance compared to those using LSTM, which implies that LSTM may be a more appropriate choice as the encoder. Given that the amount of data is relatively limited in the clinical field, it is understandable that sophisticated architecture may not outperform simple models, which explains why LSTM outperforms Transformer as the stay-level encoder. Based on the observation and reasoning, we choose LSTM instead of Transformer as the stay-level encoder in MedHMP.
>
> `>>Question 2:` __*Potential ablation study concerning stay-level pretraining*__
>
> `>>Response:` Thanks for proposing the idea of ablation study. We conduct ablating experiments regarding both patient-level and admission-level tasks, and the results are listed below.
>
>
> | Admission-level Task         |   Readmission   |
> |----------|-------|
> |           | AUROC | AUPR  |
> | Stay + MCP + CL (MedHMP) |__67.77__ | __9.34__ |
> | MCP + CL      | 65.75 | 9.08 |
> | Stay Only      | 64.87 | 8.60 |
> | Without Pretraining       | 64.74 | 8.61 |
>
>
> | Patient-level Task (LSTM) | |MIMIC             |        | |Heart Failure         |       | | COPD          |     |   | Amnesia |               |
> |--------------------------|-----------|--------|--------|---------------|--------|--------|---------|--------|--------|---------|--------|--------|
> |                          | AUPR      | F1     | KAPPA  | AUPR          | F1     | KAPPA  | AUPR    | F1     | KAPPA  | AUPR    | F1     | KAPPA  |
> | Stay + Admission (MedHMP)| __57.83__ | __59.40__ |__35.86__ | __50.16__ | __46.08__ | __29.26__ | __50.16__ | __49.34__ | __34.64__ | __48.68__ | __49.64__ | __34.46__ |
> | MCP + CL                 | 57.57     | 58.27   | 35.67   | 49.88       | 44.86   | 28.58   | 49.90    | 47.65   | 33.77   | 48.48    | 48.70   | 33.52   |
> | Without Pretraining      | __57.83__ | 56.70   | 33.03   | 48.20       | 44.44   | 26.64   | 49.52    | 47.76   | 33.44   | 47.92    | 48.80   | 32.98   |
>
> According to our experimental results, even though merely performing admission-level pretraining has already contributed to decent performance promotion, the addition of stay-level pretraing can further boost the gain and enhance model’s capability of handling downstream tasks at different levels.
>
> Besides, the ablating experiment about stay-level input during inference indicates that stay-level input is not only beneficial during training, but also indispensable during the inference, which corroborates the necessity of simultaneously taking the hierarchical input into consideration while handling EHRs.
>
>
> `>>Question 3:` __*Insufficient explanation of Masked Code Prediction (MCP)*__
>
> `>>Response:` It is our pleasure to provide more descriptions about these details. Given an admission, we randomly mask some of its ICD and drug codes (Equation 6), then fuse multimodalities via Transformer (Equation 7), these two steps allow our model to obtain the representation of admission whose part of codes are masked. Based on the representation ($a_i$), our model is expected to calculate the probability vectors ($p^c_i$, $p^g_i$).
>
> The Equation 9 is designed to urge the model to predict masked codes. We would like to sincerely thank the reviewer for pointing out the notable typo in Equation 9, which significantly introduces confusion. It should get rid of multiplication between probability and masking indicator vector (i.e. $p^c_i \odot c^m_i$ and $p^g_i \odot g^m_i$). Instead, the loss is calculated between the original output probability and masking indicator vector, which is equivalent to the representation of masked codes. The correction version of Equation 9 should be:
>
> $L_{MCP}=\frac{1}{N} \sum_{i=1}^N (||p^c_i  -  c^m_i ||_2^2 + ||p^g_i  -  g^m_i ||_2^2)$
>
> We will make the correction in our revision.
>
>
> `>>Question 4:` __*Insufficient explanation of unimodal evaluation experiments*__
>
> `>>Response:` In our setting, we add our pretrained LSTM encoder onto the top of sequential models. Basically, we combine LSTM and Transformer with the pretrained encoder in a cascade pattern rather than a transfer learning pattern where the pretrained model passes parameters to scratch models.
>
>
> `>>Question 5:` __*Absence of patient-level pretraining*__
>
> `>>Response:` As what we claimed on Page 2 footnote 3, currently we take a cautious attitude towards modeling relations between multiple admissions at the patient level. However, we admit that the interaction between admissions can be potentially modeled and we would like to continue work on that in the future.
>
> `>>Question 6:` __*Lack of novelty*__
>
> `>>Response:` Thanks for acknowledging our effort on taking the step of pretraining on fine-grained data. Admittedly, a number of colleagues have contributed a lot to the pretraining in the healthcare domain, however, hierarchical nature of EHR is relatively overlooked in previous explorations and we have taken the action of tackling this research gap with our original pretraining strategy. Besides, we have proposed in-depth analysis on the EHR hierarchy, which might be inspiring to colleagues previously omitting this research problem. We sincerely hope that our work can benefit the whole research community and we would like to continue exploring understudied areas in this direction.
>
>
> `>>Question 7:` __*Ambiguous definition of smaller-sized datasets*__
>
> `>>Response:` Thanks for pointing this out. We have provided statistics of datasets in Table 4. We will add the reference of this table in Line 383 to guide readers for a better sense of the data we use.
>
> `>>Question 8:` __*Lack of explanation of choosing two-stage pretraining strategy*__
>
> `>>Response:` Thanks for proposing the question. Reconstruction is a relatively hard objective compared to the ones we use in admission-level pretraining. While ICD codes, drug codes and other data in admission- and patient-level are relatively clean, clinical time series are highly sparse and noisy, which makes the reconstruction much more difficult than masked code prediction and contrastive learning. Based on this characteristic, we would like to separate the reconstruction from the joint pretraining for the facilitation.
>
> __Typos Grammar Style and Presentation Improvement:__
> We wholeheartedly thank the author for highlighting the typos and presentation shortcomings, and we are keen to update our manuscript in response. We would give our sincerest appreciation on the reviewer’s assistance on helping us polishing our mathematical formulas.

---

### Official Review · Reviewer_qKr6 · 2023-08-05

**Soundness:** 4

**Excitement:**

4: Strong: This paper deepens the understanding of some phenomenon or lowers the barriers to an existing research direction.

**Paper Topic And Main Contributions:**

This paper aims to leverage the hierarchical nature of electronic health records to do multi-stage pretraining. They use a reconstruction pretraining strategy to pretrain their model on the most granular, stay level. Then, they use the values that they compute in the first stage to initialize further pretraining using an intra-modality mask code prediction loss and an inter-modality contrastive learning loss. They present superior performance on a range of downstream tasks with varying time-scales.

**Questions For The Authors:**

How is a stay defined? Would be nice to clarify this more precisely in the paper. I see that you state "Each stay record includes hourly clinical monitoring readings like heart rate...". Does this mean that a stay is defined as one of the consecutive hours during an admission? Thus, the super script in S^j refers to the jth hour during an admission? Most of the information is there but would be nice to state these things a bit more precisely.

Related to the first question, on page 3, line 159, it is stated that "each stay... consists of a set of time-ordered hourly clinical features. However, I thought a stay referred to an hour during an admission? Also a subscript k is used here, but it is stated that this value k refers to the t-th hour? It may be clearer to use t instead of k here. Now I am wondering if a stay is an overnight stay? Therefore, a single admission can consist of multiple overnights?

**Reasons To Accept:**

It is clear that the authors put lots of thought into devising this hierarchical pretraining strategy and substantial work into the execution of their ideas.

It is great that the authors present a comprehensive set of experiments and tasks on various time scales.

The authors describe how the hierarchical and multi-modal nature of electronic health records can be leveraged during pretraining, which is an important contribution.

The authors make good use of published baselines that demonstrate the superiority of their method

The authors make us of compelling metrics, including AUROC, AUPR, F1 and KAPPA.

**Reasons To Reject:**

The writing and notation are a bit hard to follow at points. See the "questions for the authors" section.

There is not a comparison to a gradient boosting method. This would serve as a simple baseline that a broader audience would be familiar with. With tabular data, this method can do surprisingly well so it is important to include this.



**Reproducibility:**

4: Could mostly reproduce the results, but there may be some variation because of sample variance or minor variations in their interpretation of the protocol or method.

**Reviewer Confidence:**

4: Quite sure. I tried to check the important points carefully. It's unlikely, though conceivable, that I missed something that should affect my ratings.

**Typos Grammar Style And Presentation Improvements:**

It would be helpful to add more information to the caption of Figure 3, so that everything that is being depicted in the figure is clarified.

On page 6, line 412, what does well-trained refer to?

Would be nice to add a bit more detail to Table 3's caption.

---

> ### Author Rebuttal · Authors · 2023-08-29
>
> We are grateful for the time and effort the reviewer dedicated to understanding and commenting on our work. We summarize the concerns the reviewer holds as follows:
>
> `>>Question 1:` __*The description of “stay” is obscure*__
>
>
>  `>>Response:` We are more than willing to provide a more detailed explanation of the definition of “stay” to help readers get a better sense of the complex data structure we are aiming to tackle in this work. A simple description of the relation between admission and stay is that, admission may consist of multiple stays as the patient may be under monitored for multiple periods during the admission, while each stay includes multiple hours. In Section 2.1, we introduce the definition of S, where $S_i^j$ is the j-th stay of the patient’s i-th admission. The hourly recorded feature is defined in Section 2.2.1 in the form of $m_{i, k}^j$, which subordinates to the stay $S_i^j$. Hope the above explanation can facilitate the understanding. We do appreciate the suggestion of using t instead of k in terms of denoting the t-th hour. We will adopt this idea to further improve the readability of this study. Thanks for pointing this out!
>
> `>>Question 2:` __*Gradient boosting methods that should have been able to serve as baselines are absent*__
>
>  `>>Response:` Gradient boosting is effective for tabular data. However, fusing modalities poses challenges if we use boosting for our demographic table and DNN for other modalities. This is because traditional boosting doesn't produce representations suitable for fusion. We could consider applying boosting to all modalities. Yet, for time series and clinical texts, boosting might not capture sequential nuances effectively. Given these considerations, we feel omitting gradient boosting in a multimodal context is justifiable. Its absence only marginally impacts the study's solidness and rigor. Nonetheless, we're grateful to the reviewer for initiating this insightful discussion on a potential direction.
>
> __Typos Grammar Style and Presentation Improvement:__
>
> `>>1.` __*Absence of clear definition of “well-trained”*__
>
>  `>>Response:` Here, the word “well-trained” refers that we have performed two-stage pretraining on the LSTM encoder, as the counterpart of the one merely trained at stay level. We thank the reviewer for proposing the doubt and we would like to add more descriptions about this part in our camera-ready version of paper.
>
> We truly value the author's observations and suggestions regarding Figure 3, and we plan to make the adjustments accordingly to the final version of our manuscript.

---

### Official Review · Reviewer_P82s · 2023-08-19

**Typos Grammar Style And Presentation Improvements:** N/A
**Soundness:** 4

**Excitement:**

4: Strong: This paper deepens the understanding of some phenomenon or lowers the barriers to an existing research direction.

**Missing References:**

N/A

**Paper Topic And Main Contributions:**

Paper Topic: Electronic Health Records, Multi-modal Representation Learning

Main Contribution:

This paper aims to perform representation learning for Electronic Health Records, which normally contain three levels: patient, admission-level, and stay-level.
* For stay-level pre-training, it adopts Feature Reconstruction as its pre-training task with the consideration of demographic information.
* For admission-level pre-training, there are two types of objectives: one is to predict the masked codes and the other is to perform contrastive learning between original features and noisy features.
After such reasonable representation learning, the trained model can achieve promising results in different levels of evaluation.

**Questions For The Authors:**

* Regarding the source code: I wonder if the authors will make public the code, including the pre-processing, training, and evaluation.

**Reasons To Accept:**

* The paper is well-written. I enjoyed reading it and also learned a lot from it.
* From the perspective of representation learning, the proposed approach is designed for different data types in an appropriate way. Specifically, the authors use reconstruction for numerical data, classification for the discrete set, and contrastive learning from cross-modal data.
* The evaluation is comprehensive, across patient-level, admission-level, and stay-level evaluation. Besides, the proposed representation learning works well in all the evaluations.

Overall, I think this is an excellent paper and I will recommend the acceptance.

**Reasons To Reject:**

One thing I am curious about except for the downstream evaluation is the analysis of the pre-training analysis. To be specific, there are too many variables in the representation learning process and I wonder about the difficulty of different pre-training tasks. For example, some tasks might be easy since the model finds the shortcut and some might be key to the downstream performance.

**Reproducibility:**

4: Could mostly reproduce the results, but there may be some variation because of sample variance or minor variations in their interpretation of the protocol or method.

**Reviewer Confidence:**

4: Quite sure. I tried to check the important points carefully. It's unlikely, though conceivable, that I missed something that should affect my ratings.

---

> ### Author Rebuttal · Authors · 2023-08-29
>
> The insights and expertise displayed by the reviewer in their feedback are truly invaluable to us. We summarize the concerns that the reviewer raise as follows:
>
> `>>Question 1:` __*Absent ablation study on different pretraining tasks*__
>
>  `>>Response:` Thanks for proposing the concern. We have conducted experiments to investigate the usefulness of different pretraining tasks in MedHMP, and results can be found below:
>
> |          |   ARF  ||     Shock     ||    Mortality   ||
> |----------|-------|-------|-------|-------|--------|-------|
> |          | AUROC | AUPR  | AUROC | AUPR  | AUROC  | AUPR  |
> | MedHMP | __71.66__ | __14.34__ | __71.04__ | __24.19__ | __82.17__  | __47.52__ |
> | Stay + MCP      | 64.91 | 12.35 | 68.61 | 25.29 | 81.32  | 47.50 |
> | Stay + CL       | 62.99 | 13.88 | 70.05 | 22.81 | 80.58  | 44.40 |
> | Stay Only | 64.65  | 10.59  | 67.94  | 22.50  | 79.67   | 42.66  |
>
> According to our experiments, the combination of them manifest best performance, thus each pretraining task has been proven to be indispensable. As for the two objectives that we jointly use in admission-level pretraining, MCP appears to be slightly more effective. However, it cannot independently help the model achieve the best performance. Therefore, we would like to claim that each pretraining task can be regarded as key to the downstream performance.  We will add these experimental results to our camera-ready version of this study that serves as a more comprehensive analysis on our pretraining strategy.
>
> `>>Question 2` __*Question about open-source option*__
>
>  `>>Response:` We have provided our source codes including training and evaluation in  the supplemental materials, as well as guidance on how to adapt existing pre-processing pipeline. We will prepare and release a more detailed version of these resources after the acceptance of this work.

---

### Official Review · Reviewer_TZeS · 2023-08-19

**Soundness:** 4

**Ethical Concerns:**

Yes

**Excitement:**

3: Ambivalent: It has merits (e.g., it reports state-of-the-art results, the idea is nice), but there are key weaknesses (e.g., it describes incremental work), and it can significantly benefit from another round of revision. However, I won't object to accepting it if my co-reviewers champion it.

**Justification For Ethical Concerns:**

This paper employs a dataset from TriNetX, which is not open-sourced, for the validation of certain experiments. This raises several potential ethical concerns that warrant a separate review:
- The absence of an explicit section or statement addressing ethical considerations in the paper indicates a lack of reflection on these crucial aspects.
- The use of a non-open-sourced dataset may involve patient data or other sensitive information. The lack of transparency and details about the data collection, consent procedures, and privacy safeguards can raise questions about the ethical handling of potentially sensitive information.
- https://trinetx.com/real-world-resources/publications/trinetx-publication-guidelines/ refers to the publication guidelines of TriNetX. A review should be conducted to ensure that the researchers have complied with these guidelines, including any requirements for data use, acknowledgment, or disclosure.

**Paper Topic And Main Contributions:**

This paper addresses the challenge of combining the hierarchical (i.e., patient-admission-stay) and heterogeneous (i.e., multimodal or varied in data type) nature of EHR data into one pretraining model. This model aims to handle diverse medical predictive tasks, covering eight downstream tasks over three levels.

The main contributions include:
- A new two-stage ("bottom-to-up") hierarchical multimodal pretraining framework specifically designed for EHR data.
- The development of level-specific self-supervised learning tasks, including (1) stay-level reconstruction, (2) admission-level intra-modality modeling, and (3) admission-level inter-modality modeling.
- Comprehensive experiments on diverse clinical benchmarks across two databases, demonstrating the superior performance of MedHMP across various tasks and outperforming eighteen baseline models.

**Questions For The Authors:**

- **Question A**: For most experiments, several simple but effective baselines (with the same size of parameters) could be included to show the efficacy of pre-training: 1) the fine-tuned model without any pretraining; 2) the fine-tuned model with only stay-level pretraining ($L_{stay}$); 3) the fine-tuned model with only admission-level pretraining ($L_{MCP} + L_{CL}$). Have you conducted these baselines for most experiments?
- **Question B**: Do we really need two objectives in the admission-level pretraining? Have you conducted ablation studies for intra-/inter-modalities? For example, compare four types of experiments such as (1) $L_{stay} \rightarrow L_{MCP} + L_{CL}$ vs. (2) $L_{stay} \rightarrow L_{CL}$ vs. (3) $L_{stay} \rightarrow L_{MCP} $ vs. (4) only $L_{stay}$
- **Question C**: There are several possible admission-level tasks, but only one is presented here. Could you explain the reasoning behind this choice?
- **Question D**: In the "stay-level unimodal evaluation" section, could you clarify how this experiment specifically contributes to validating the generalizability of MedHMP? (e.g., what aspects of the evaluation are designed to test generalizability, and how do the results support this claim?)
- **Question E**: I am concerned about the generalization of this approach, as it is based on medical codes. Could this pre-trained model be applied to other medical databases such as eICU?
- **Question F**: On Line 250, the term "intra-modality" may be causing confusion, especially in the context of the model's use of admission representation by fusing different modalities. This confusion was heightened for me when reading the context in Lines 291-293. Could you elaborate on the definitions of "intra-modality" and "inter-modality"?
- **Question G**: What is the motivation of using other private clinical benchmark dataset (from TriNetX)?
- **Question H**: Considering that this work is specific to the EHR domain, even though it tackles the multi-modal pre-training scheme, what would be the main benefits of this work to the general NLP community, and how might it also be beneficial for the clinical NLP community?

**Reasons To Accept:**

- MedHMP is a novel framework for multimodal EHRs, addressing gaps with hierarchical and heterogeneous data.
- It spans patient, admission, and stay levels, for broad applicability in predictive healthcare.
- Source code inclusion enhances transparency and reproducibility.
- Focus on hierarchical EHR data may inspire development in domain-specific pretraining models beyond healthcare.

**Reasons To Reject:**

- Insufficient experimental designs/results (Please refer to the "Questions For The Authors" section.)
- Insufficient justification for pre-training decisions (Please refer to the "Questions For The Authors" section.)
- Not sure for generalizability (Please refer to the "Questions For The Authors" section.)
- Potential ethical or privacy considerations (Please refer to the "Justification For Ethical Concerns" section.)

**Reproducibility:**

4: Could mostly reproduce the results, but there may be some variation because of sample variance or minor variations in their interpretation of the protocol or method.

**Reviewer Confidence:**

4: Quite sure. I tried to check the important points carefully. It's unlikely, though conceivable, that I missed something that should affect my ratings.

**Typos Grammar Style And Presentation Improvements:**

- L134: Please specify the demographic features used in the model and provide an example.
- L138: Please specify the types of notes considered (e.g., discharge summary).
- L143: Please consider the revision of notation from $M$ to $M_i$ for dependency on admission $i$.
- L231: Please add the context how the model handles varying lengths of $M$ (e.g., padding).
- L240-242: Please specify clinical notes types and explain text processing if too lengthy (e.g., maximum sequence length).
- L272: Please clarify dimensions of $f_i$; if 4*$d_r$, state explicitly and correct the manuscript.
- L333: Please include standard deviations in results for full statistical context.
- Table 3: Please correct one bold formatting error.
- L383: Please define "smaller-sized datasets."
- L472: Please provide information on AUROC as well as AURPC.
- L982: Please check context of "Table 6" in admission-level task section; clarify if needed.
- Figures 1/2/3: Please ensure captions are self-explanatory.

---

> ### Author Rebuttal · Authors · 2023-08-29
>
> We deeply appreciate the reviewer’s invaluable comments and constructive suggestions, which significantly improve the quality of our paper. We hope the responses below can adequately address the reviewer’s concerns.
>
> `>>Question A:` __*Efficacy of pre-training at different levels.*__
>
> `>>Response:`Thanks for pointing this out. To validate the performance of the three approaches mentioned by the reviewer, we conduct the following three experiments at different levels.
>
> We first compare the three approaches mentioned by the reviewer with the proposed MedHMP on the admission-level task - readmission prediction, since this level task can validate the three baselines simultaneously. We then use the tasks at the stay level to exhibit the performance of the models without pre-training and using stay-level pre-training only. Finally, we use the patient-level task - risk prediction - to show the performance of the models without pre-training and using the admission-level pre-training only. The results are shown in the following three tables:
>
>
> | Admission-level Task         |   Readmission   | |
> |----------|-------|-------|
> |           | AUROC | AUPR  |
> | Stay + MCP + CL (MedHMP) | __67.77__  | __9.34__ |
> | MCP + CL      | 65.75 | 9.08 |
> | Stay Only      | 64.87 | 8.60 |
> | Without Pretraining       | 64.74 | 8.61 |
>
>
> |Stay-level Tasks          |   ARF  ||     Shock     ||    Mortality   | |
> |----------|-------|-------|-------|-------|--------|-------|
> |          | AUROC | AUPR  | AUROC | AUPR  | AUROC  | AUPR  |
> | MedHMP | __71.66__ | __14.34__ | __71.04__ | __24.19__ | __82.17__  | __47.52__ |
> | Stay Only | 64.65  | 10.59  | 67.94  | 22.50  | 79.67   | 42.66  |
> | Without Pretraining | 64.06  | 10.80 | 67.71  | 23.19  | 79.04   | 40.12  |
>
> | Patient-level Task (LSTM) | |MIMIC             |        | |Heart Failure         |       | | COPD          |     |   | Amnesia |               |
> |--------------------------|-----------|--------|--------|---------------|--------|--------|---------|--------|--------|---------|--------|--------|
> |                          | AUPR      | F1     | KAPPA  | AUPR          | F1     | KAPPA  | AUPR    | F1     | KAPPA  | AUPR    | F1     | KAPPA  |
> | Stay + Admission (MedHMP)| __57.83__ | __59.40__ |__35.86__ | __50.16__ | __46.08__ | __29.26__ | __50.16__ | __49.34__ | __34.64__ | __48.68__ | __49.64__ | __34.46__ |
> | MCP + CL                 | 57.57     | 58.27   | 35.67   | 49.88       | 44.86   | 28.58   | 49.90    | 47.65   | 33.77   | 48.48    | 48.70   | 33.52   |
> | Without Pretraining      | __57.83__ | 56.70   | 33.03   | 48.20       | 44.44   | 26.64   | 49.52    | 47.76   | 33.44   | 47.92    | 48.80   | 32.98   |
>
>
> From the results of all the three tables, we can observe that the proposed MedHMP outperforms those baselines consistently, which demonstrates the efficacy of the proposed pre-training strategy. Besides, using a part of the proposed pre-training strategy can improve the performance, compared with the model without any pre-training techniques. We will add these results to the final version.
>
> `>>Question B:` __*Lack of ablation study on different components of admission-level loss.*__
>
> `>>Response:`Thanks for the comment. To validate the necessity of these two losses in our admission-level pre-training, we conduct the following experiment as the reviewer suggested. We take the three tasks at the stay-level as an example. The experimental results are shown in the following table. We can observe that using either loss term proposed at the admission level pre-training on top of the stay-level pre-training, the performance is better than that only using the stay-level pre-training. Besides, using them together can further enhance the performance. These results clearly demonstrate the efficacy of the proposed loss terms and the designed pre-training strategy. We will add these results to the final version.
>
> |          |   ARF  ||     Shock     ||    Mortality   ||
> |----------|-------|-------|-------|-------|--------|-------|
> |          | AUROC | AUPR  | AUROC | AUPR  | AUROC  | AUPR  |
> | Stay + MCP + CL (MedHMP) | __71.66__ | __14.34__ | __71.04__ | __24.19__ | __82.17__  | __47.52__ |
> | Stay + MCP      | 64.91 | 12.35 | 68.61 | 25.29 | 81.32  | 47.50 |
> | Stay + CL       | 62.99 | 13.88 | 70.05 | 22.81 | 80.58  | 44.40 |
> | Stay Only | 64.65  | 10.59  | 67.94  | 22.50  | 79.67   | 42.66  |
>
> `>>Question C:` __*Absence of more admission-level tasks*__
>
> `>>Response:`Thanks for pointing this out. In our experiments, we pre-train the proposed model on the MIMIC datasets. For the admission-level tasks on the MIMIC datasets, hospital readmission prediction is a representative task and has been widely explored by researchers. Another commonly-used task is diagnosis prediction (Yang and Wu, 2021), where ICD codes are used as labels. However, the mask code prediction strategy adopted in our pretraining procedure also takes ICD codes as input. To avoid the issue of label leakage, we do not list this task in our experiments. We will explore other admission-level downstream tasks in our future work.
>
> `>>Question D:` __*Ambiguity of generalizability*__
>
> `>>Response:`Thanks for the comment. The experiments of the stay-level unimodal evaluation aim to demonstrate the efficacy of the pre-trained encoder, which can be used to further enhance the performance of  original baselines. The word “generalizability” is confusing, and we will rephrase the sentence in lines 397-400.
>
> `>>Question E:` __*Concern about generalization on other medical databases*__
>
> `>>Response:`Please note that medical codes are a specific modality in the pre-training. From the results listed in Table 1 in our original paper and Questions A and B, we can observe that even for the tasks without using medical codes, the designed pre-training strategy still performs better than baselines, since MedHMP can pre-train on the stay-level data. To validate the generalization of our model, we extract a new dataset from eICU to predict mortality after 48 hours. The experimental results are shown in the following table. The results still demonstrate the efficacy of the proposed MedHMP on this new dataset. We will add these results to the final version.
>
> |          |    Mortality   ||
> |----------|--------|-------|
> |          | AUROC  | AUPR  |
> | F-LSTM | 82.30  | 45.01  |
> | F-CNN | 76.37   | 35.11  |
> | DCMN       | 83.64   | 46.40  |
> | RAIM       |  83.57  | 46.96  |
> | MedHMP | __84.43__   | __49.00__  |
>
> `>>Question F:` __*Unclear definitions of intra- and inter-modality*__
>
>  `>>Response:` Thanks for pointing this out. Intra-modality modeling aims to capture the feature relations within a specifc modality, such as ICD codes and medication codes, with the proposed masked code prediction. For example, assume that the total number of ICD codes is 5 in a dataset. For an admission, its ICD code binary vector is [1, 0, 1, 1, 0]. Assume that we randomly mask one code, which results in a masked ICD code vector [1, 0, 1, MASK, 0]. Our goal is to predict whether the masked position is 1 or 0.
>
> Inter-modality interaction tries to model the relation cross modalities. For instance, a list of ICD codes should have a strong connection with clinical notes that belong to the same admission, while clinical notes for other admission may not be highly irrelevant. By adopting our proposed contrastive learning loss, the model acquires the capability of capturing correspondence between different modalities, hence better estimate the data distribution across modalities.
>
> We will clearly elaborate these two terms in the final version to avoid any confusion.
>
> `>>Question G:` __*Motivation of adopting private datasets*__
>
>  `>>Response:` Even though we extract a heart failure dataset from the MIMIC-III database in our experiment, the recorded data in the MIMIC-III database are from ICU patients, which is too specific to reflect the characteristics of EHR data. Besides, the average number of admissions for each patient is very small, which is only 2.61. Finally, all the tasks used in the experiments at the stay and admission levels are extracted from the MIMIC databases. In other words, the pre-training and downstream tasks are validated using the data with the same distribution.
>
> To validate the proposed MedHMP in a more general scenario, we extract three new datasets from the TriNetX database. The EHR data in these datasets record any types of admissions, not only limited to the ICU ones, and the average number of admissions per patient is more than 30. We believe validation on these private datasets can confirm the generalization ability and efficacy of the proposed model.
>
> `>>Question H:` __*Inquiry about broad impact on (Clinical) NLP community*__
>
> `>>Response:` Thank you for posing the insightful question. We posit that our emphasis on integrating the inherent hierarchical nature of data within the pretraining process has broad applicability across the NLP spectrum. Consider semi-structured data like web content, which inherently exhibits a hierarchical structure often intertwined with multiple modalities. Traditional NLP models pretrained on generic corpora might struggle with the noisy text without utilizing additional information. Our pretraining approach, which is capable of leveraging both hierarchical and multimodal attributes, can serves as a beacon for NLP researchers aiming to pretraining their model similarly for a decent adaptability.
>
> For the clinical NLP community, our work provides a direction that has not been fully explored by previous studies, which fills a research gap where EHR hierarchy has not been considered while pretraining clinical NLP models. We believe that clinical NLP researchers will be inspired by our current exploration, and practitioners are enabled to acquire more effective pretrained language models for their application demands.
>
> __Ethical Concern:__
>
> `>>Response:` The data in the TriNetX database are de-identified and do not contain any sensitive patient information. We have obtained the access to this database within our institute. Thus, we do not think there are potential ethical concerns when using this database.
>
> __Typos Grammar Style and Presentation Improvement:__
>
> `>>1.` __Demographic features:__
>
> `>>Response:` Following the previous work (Tang et al., 2020), we extract and utilize the following demographic features: gender, admission type, language, age, wardid (identifier of ward), admission location, marital status, religion, care unit, ethnicity, and insurance type. An example is listed below:
>
> |      Feature          | Value                       |
> |----------------|-----------------------------|
> | Gender         | F                           |
> | Admission type | EMERGENCY                     |
> | Language       | SPAN                        |
> | Age            | 59                          |
> | Wardid | WARDID=50                     |
> | Admission location | CLINIC REFERRAL/PREMATURE     |
> | Marital status | MARRIED                     |
> | Religion | CATHOLIC                    |
> | Care Unit| MICU                      |
> | Ethnicity      | HISPANIC/LATINO - DOMINICAN |
> | Insurance| Medicare |
>
> `>>2.` __Type of notes:__
>
> `>>Response:` Still following the previous work (Yang and Wu, 2021), we leverage a wide range of clinical notes including not only discharge summary, but also pharmacy notes, nursing notes, physician notes, radiology reports, ECG reports, echo reports, respiratory notes, nutrition notes, rehabilitation notes, social work notes and case management notes in the pretraining and downstream tasks. The inclusion of diverse types of clinical notes ensures that a fair comparison can be performed between our method and baselines mentioned in this work.
>
>
> `>>3.` __Subscript of M:__
>
> `>>Response:` Thanks for detecting the omission. We will make the modification in our camera-ready version.
>
> `>>4.` __How the model handles various lengths of M:__
>
> `>>Response:` We adopt the padding operation to ensure the consistency of the number of stays across different admissions. According to our statistics, only less than 2% of admissions possess more than 3 stays. Therefore, we set the maximum number of stays to 3.
>
>
> `>>5.` __Setting of maximum sequence length:__
>
> `>>Response:` Similar to most Transformer-based language models, the maximum sequence length of Clinical-T5 is set to 512 by default. Moreover, we adopt the preprocessing step suggested by the provider of ClinicalT5 (https://physionet.org/content/clinical-t5/1.0.0/) to reduce the length and noise of clinical text.
>
>
> `>>6.` __Dimension of $f_i$:__
>
> `>>Response:` The operation that we take to aggregate representations of different modalities is vertical stacking rather than horizontal concatenation. Therefore, the dimension of $f_i$ is still $d_r$, while the representation size of each data sample changes from 1*$d_r$ to 4*$d_r$ after the aggregation.
>
> `>>7.` __About standard derivation:__
>
> `>>Response:` Thanks for proposing the addition. We will make modifications accordingly in our final version of this study.
>
>
> `>>8.` __Bold error in Table 3:__
>
> `>>Response:` Thanks for pointing this out. We will correct this error.
>
> `>>9.` __Smaller-sized datasets:__
>
> `>>Response:` The size of datasets are described in Table 4, based on which we claim that ARF and Shock datasets are smaller in size. We will add the reference of Table 4 to Line 383 to guide readers for a better sense of the data we use.
>
> `>>10.` __Definition of AUROC and AUPRC:__
>
> `>>Response:` AUROC measures the overall ability of a model to discriminate between the positive class and the negative class across different decision thresholds. AUPR measures the model's precision and recall across different decision thresholds, especially in the context of imbalanced datasets. We will add their clear definitions to the final version of our paper as well as the definitions of other evaluation metrics (F1, KAPPA) we use in our experiments.
>
>
> `>>11.` __Context of Table 6:__
>
> `>>Response:` Previous work (Yang and Wu, 2021) has their experiments conducted in a different setting, where each admission only corresponds to one stay. In this setting, the admission- and stay-level tasks are mixed up with each other. We argue that this setting ignores the hierarchical nature of EHRs, therefore, we only leverage stay-level information in our experiments regarding ARF, shock and mortality predictions as we treat them as stay-level tasks, and choose readmission prediction as our admission-level tasks, where multiple stays serve as input for each admission.
>
> The reason we conduct experiments in Table 6 is that we would like to avoid any omission of possible baselines, even if their setting is different from ours. Hence, we include Table 6 in the appendix, which further corroborates the efficacy of MedHMP in contrast to baselines. Moreover, the comparison between Table 6 and Table 2 indicates that our model does not manifest better performance given additional input, validating our assumption that mixing up stay- and admission-level tasks may cause problems due to the mismatching information in different granularities.
>
>
> `>>12.` __Captions improvement:__
>
> `>>Response:` Thanks for proposing the problem. We will add more detailed captions in our final version to ensure the self-containedness of our work.
>
> We sincerely appreciate the typos and presentation problems the reviewer has pointed out, and we would like to make modifications on our manuscript accordingly. We will add the clarification above to our paper to improve both rigor and readiness of this work based on your valuable suggestion.

---

### Meta-Review · Area_Chair_8Lhe · 2023-09-19

**Recommendation:** 5

**Metareview:**

The paper presents a two-stage hierarchical multimodal pretraining framework, named MedHMP, which is designed for Electronic Health Records (EHR) data. This framework aims to address the challenges of managing the hierarchical and heterogeneous nature of EHR data. By performing extensive experiments across diverse clinical benchmarks, the paper convincingly demonstrates that MedHMP outperforms existing baseline models. The hierarchical multimodal approach is well-suited for EHR data, offering a promising direction for predictive healthcare. The reviewers largely agree that the study provides strong support for its claims. Given that the majority of the reviewers found the work both sound and exciting, recommending a strong acceptance seems justified. However, the authors should pay close attention to the detailed questions and concerns raised by the reviewers in future revisions.

---

### Decision · Program_Chairs · 2023-10-07

**Decision:**

Accept-Main

**Comment:**

The paper presents a two-stage hierarchical multimodal pretraining framework, named MedHMP, which is designed for Electronic Health Records (EHR) data. This framework aims to address the challenges of managing the hierarchical and heterogeneous nature of EHR data. By performing extensive experiments across diverse clinical benchmarks, the paper convincingly demonstrates that MedHMP outperforms existing baseline models. The hierarchical multimodal approach is well-suited for EHR data, offering a promising direction for predictive healthcare. The reviewers largely agree that the study provides strong support for its claims. Given that the majority of the reviewers found the work both sound and exciting, recommending a strong acceptance seems justified. However, the authors should pay close attention to the detailed questions and concerns raised by the reviewers in future revisions.